# Association of Inflammation, Ectopic Bone Formation, and Sacroiliac Joint Variation in Ossification of the Posterior Longitudinal Ligament

**DOI:** 10.3390/jcm12010349

**Published:** 2023-01-02

**Authors:** Nguyen Tran Canh Tung, Zhongyuan He, Hiroto Makino, Taketoshi Yasuda, Shoji Seki, Kayo Suzuki, Kenta Watanabe, Hayato Futakawa, Katsuhiko Kamei, Yoshiharu Kawaguchi

**Affiliations:** 1Department of Orthopaedic Surgery, Faculty of Medicine, University of Toyama, Toyama 930-0194, Japan; 2Department of Trauma and Orthopaedic Surgery, Vietnam Military Medical University, Hanoi 100000, Vietnam; 3Innovation Platform of Regeneration and Repair of Spinal Cord and Nerve Injury, Department of Orthopaedic Surgery, The Seventh Affiliated Hospital of Sun Yat-sen University, Shenzhen 518107, China

**Keywords:** ossification of the posterior longitudinal ligament, hs-CRP, sacroiliac joint, inflammation

## Abstract

Ossification of the posterior longitudinal ligament (OPLL) is considered a multifactorial condition characterized by ectopic new bone formation in the spinal ligament. Recently, its connections with inflammation as well as sacroiliac (SI) joint ankylosis have been discussed. Nevertheless, whether inflammation, spinal ligament ossification, and SI joint changes are linked in OPLL has never been investigated. In this study, whole-spinal computed tomography and serum high-sensitive C-reactive protein (hs-CRP) levels were obtained in 162 patients with cervical OPLL. Ossification lesions were categorized as plateau and hill shapes. Accordingly, patients were divided into plateau-shaped (51 males and 33 females; mean age: 67.7 years) and hill-shaped (50 males and 28 females; mean age: 67.2 years) groups. SI joint changes were classified into four types and three subtypes, as previously described. Interactions among ossification shapes, hs-CRP levels, and morphological changes in the SI joint were investigated. The plateau shape was more common in the vertebral segments (59.5%), compared to the hill shape, which was predominant in the intervertebral regions (65.4%). Serum hs-CRP levels in the plateau-shaped group (0.11 ± 0.10 mg/dL) were significantly higher than those in the hill-shaped group (0.07 ± 0.08 mg/dL). SI joint intra-articular fusion was the main finding in the plateau-shaped group and showed significantly higher hs-CRP levels compared to the anterior para-articular bridging, which more frequently occurred in the hill-shaped group. Our findings suggested a possible inflammation mechanism that might contribute to the new bone formation in OPLL, particularly the plateau shape.

## 1. Introduction

Ossification of the posterior longitudinal ligament (OPLL) is characterized by the ectopic bone formation that occurs in the PLL due to hyperostotic changes [1]. The progression of these ossified lesions may lead to neural compression, which can cause serious symptoms such as myelopathy and/or radiculopathy that require clinician attention. OPLL pathogenesis is multifactorial and remains to be fully elucidated. Consequently, there is no disease-modifying treatment, and thus, clinical interventions are mainly focused on symptom relief, with severe cases managed through surgical treatment to relieve spinal cord compression.

OPLL has been related to several other entities, such as diffuse idiopathic skeletal hyperostosis (DISH) and ankylosing spondylitis (AS) [2,3,4,5]. DISH and OPLL are most frequently characterized by degenerative changes without the appearance of sacroiliitis [6,7,8], while AS is a common chronic inflammatory rheumatic disease that starts in the sacroiliac (SI) joints and spreads to the spine [9,10]. The radiographic appearance of both diseases is very similar, but they have different underlying pathologies. Recently, two types of new bone formation in the anterior longitudinal ligament (ALL) in DISH, as well as AS, have been reported [11,12,13,14]. One is thin, flat, vertical, and prevalent in patients with AS; the other is thick, jaggy, horizontal, and more common in patients with DISH. Our previous study classified new bone formation in the ALL in DISH into flat and jaggy types [15]. The jaggy-type ossification is caused by a degenerative process, while the flat-type might be due to an inflammatory process similar to the “syndesmophyte” pattern in AS. Interestingly, these two types of new bone formation are also observed in OPLL and have been classified into plateau and hill shapes by Iwasaki et al. [16]. However, the author mainly focused on the surgical strategies rather than providing details or possible mechanisms of the lesions themselves.

Multiple studies have looked into associated factors that contribute to ossification development in OPLL. A close relationship between OPLL, as well as OPLL progression, and high-sensitivity C-reactive protein (hs-CRP) has been proposed, suggesting an inflammatory mechanism might be related to OPLL pathogenesis [17]. The pathological bone formation has also occurred in the SI joint, causing SI joint ankylosis in two forms. One is SI joint fusion through the auricular surface (intra-articular), which is characterized by articular inflammation and considered a manifestation of AS [10]. The other is SI joint bridging but separate from the auricular component of the joint (para-articular), which was thought to be associated with DISH [18]. However, previous studies that reported the presence of SI joint intra-articular ankylosis in patients with DISH and those with OPLL [19,20] may suggest that the diseases have a pathogenetic route similar to AS. Although there is no doubt that spinal inflammation, SI joint intra-articular fusion, and new bone formation both occur in AS [10,12], less is known about OPLL in this regard. Since we know that occurrence of flat-type bony fusion in DISH might be related to inflammation, we hypothesize that the plateau shape of new bone formation in OPLL might also be related to inflammation in the spinal column and perhaps inflammation elsewhere in the SI joint.

The present study took advantage of the previous classification approach [16] and classified the new bone formation in OPLL into “plateau-shaped” and “hill-shaped.” Further, the characteristics of these two patterns and related factors, including serum hs-CRP levels and SI joint ankylosis, were investigated to elucidate the possible mechanisms that might promote the new bone formation in OPLL.

## 2. Materials and Methods

### 2.1. Patient Selection

This study was approved by the Ethics Committee at the Toyama University Hospital (approval number R2015003). All patients provided standard written informed consent to participate in the study. 162 patients who were admitted to the Department of Orthopedic Surgery at Toyama University Hospital, Japan, from 2012 to 2014 were enrolled. Patients with comorbid cardiovascular diseases were excluded based on clinical examinations and medical histories.

### 2.2. Measurement of Serum hs-CRP Level

Peripheral blood samples were collected from all patients on the morning of their hospital visit, and the serum was immediately frozen at −80 °C until analysis. An ultrasensitive latex-enhanced immunoassay was obtained to measure the serum hs-CRP concentrations as previously described, using a BN ProSpec nephelometer (Dade Behring, Newark, NJ, USA) [17]. Results were expressed as milligrams per deciliter (mg/dL).

### 2.3. Radiographic Assessment

The OPLL diagnosis was based on plain radiograph observation and computed tomography (CT) imaging of the whole spine.

The ectopic bone formation shape in the PLL was categorized into plateau and hill shapes using sagittal-view CT and based on the approach of Iwasaki et al. [16] (Figure 1). Briefly, the relatively narrow spinal canal without any localized massive ossification was assumed to present the plateau-shaped bone formation. Contrastingly, the hill-shaped ossified pattern is defined by a massive beak-shaped ossification that is localized to certain levels. Accordingly, patients were divided into two groups as follows: plateau-shaped and hill-shaped groups. The characteristics of these two patterns were assessed as per convention in the literature [21], including the maximum ossified PLL thickness and its level, the space available for the spinal cord (SAC), and the occupying ratio (OR) of OPLL. OPLL types were also classified as continuous, segmental, mixed, or localized according to the Japanese Investigation Committee for Ossification of the Spinal Ligament criteria [22].

Morphological SI joint changes were categorized into four types based on the previously reported criteria as follows [19,20]: type 1: normal or small peripheral bone irregularity; type 2: subchondral bone sclerosis and osteophyte formation; type 3: the vacuum phenomenon; and type 4: bridging osteophyte and bony fusion. Type 4 was further divided into three subgroups as follows: type 4A: anterior para-articular bridging; type 4B: posterior para-articular bridging; and type 4C: intra-articular ankylosis.

Two orthopedic surgeons independently evaluated the bone formation shapes and SI joint changes. Different opinions are discussed by the two examiners before making the final decision. Inter- and intra-observer agreements of the OPLL shape classification were calculated as intraclass correlation coefficients (ICCs) using 20 randomly selected patients.

### 2.4. Data Analysis and Statistics

Serum hs-CRP levels were compared among two OPLL shape groups and four OPLL types, and the occurrences of these shapes and types were evaluated. Furthermore, the interactions between serum hs-CRP levels, SI joint changes, and OPLL shapes were investigated.

Data were presented as the mean ± standard deviation. The differences in the quantitative data were tested by the Student’s unpaired t-test and the one-way analysis of variance, followed by Tukey’s post-hoc test. Categorical data differences were measured by the chi-squared test. Statistical analysis was conducted with GraphPad Prism v9.0.1 software (GraphPad Software, San Diego, CA, USA) and Excel statistical software (Statcel version 4; OMS, Tokorozawa, Japan). Post hoc power analyses were conducted using G*Power version 3.1.9.6. *p*-values of <0.05 were considered statistically significant. ICC values were categorized as poor (0.00–0.20), fair (0.21–0.40), moderate (0.41–0.60), strong (0.61–0.80), and almost perfect agreements (0.81–1.00).

## 3. Results

### 3.1. Baseline Characteristics

The baseline characteristics of enrolled patients are summarized in Table 1. Of the 162 patients with cervical OPLL, the plateau-shaped group was slightly predominant with 84 patients (51 males and 33 females; mean age: 67.7 years), whereas the hill-shaped group consisted of 78 patients (50 males and 28 females; mean age: 67.2 years). No significant difference was found in the demographic data between the two groups.

### 3.2. Morphological Characteristic of Ectopic Bone Formation in OPLL

The OPLL shape evaluation revealed strong inter-observer agreement (ICC = 0.796, *p* < 0.001) and almost perfect intra-observer agreement (ICC 1 = 0.900, *p* < 0.001; ICC 2 = 0.813, *p* < 0.001), indicating substantial reliability.

The ossified lesion characteristics are also presented in Table 1. The plateau shape showed a lower maximum thickness, OR, and a higher SAC than the hill shape. The ectopic new bone formation was seen at both vertebral and intervertebral levels. Interestingly, the vertebral segments were more affected than the intervertebral segments in the plateau-shaped group, with a prevalence of 59.5%. Conversely, the maximum thickness of ossified lesions occurred mainly in the intervertebral sites in the hill-shaped group (65.4%).

Figure 2 shows the relationship between OPLL shapes and types, wherein the OPLL shapes differ among the types. The more prevalent plateau shape was included in the continuous and segmental types, while the more predominant hill shape was found in the mixed and localized types.

### 3.3. The Associations between the OPLL Shapes, SI Joint Variation, and hs-CRP Levels

The associations between the OPLL shapes, SI joint variation, and hs-CRP levels were also investigated. The mean hs-CRP concentration was 0.11 ± 0.10 mg/dL in the plateau-shaped group and 0.07 ± 0.08 mg/dL in the hill-shaped group, yielding a statistical difference between groups (*p* = 0.004) (Figure 3A). Additionally, patients with the continuous type had significantly higher hs-CRP levels compared to patients with the localized type (*p* = 0.03) (Figure 3B).

All patients with OPLL presented SI joint changes, such as osteophyte formation (Type 2), SI joint vacuum phenomenon (Type 3), and SI joint fusion (Type 4). No significant differences were found in the hs-CRP levels between patients with four types of SI joint variation (Figure 4A). However, the hs-CRP levels were significantly higher in patients with SI joint intra-articular fusion (Type 4C) compared to patients with SI joint anterior para-articular bridging (Type 4A) (*p* = 0.04) when compared between SI joint subtypes (Figure 4B).

Consistent with the hs-CRP criteria, statistical differences were found between the SI joint ankylosis and OPLL shapes (Table 2). SI joint intra-articular fusion (Type 4C) was frequently detected in the plateau-shaped group (62.2%). Conversely, the SI joint anterior bridging was the most common change in the hill-shaped group (57.1%).

The post hoc power of the student’s unpaired t-test for the comparison of serum hs-CRP levels between the OPLL shapes (calculated effect size of 0.44, α = 0.05, total sample size of 162) was 0.80. The post hoc power of the one-way analysis of variance for the comparison of serum hs-CRP levels between SI joint ankylosis (calculated effect size 0.30, α = 0.05, total sample size 79) was 0.85. The post hoc power of the chi-squared test for the association between OPLL shapes and SI joint ankylosis (calculated effect size of 0.68, α = 0.05, total sample size of 79) was 0.999. Thus, we suggest that the modest sample size in the present study was sufficient to detect the significant interaction effects in our findings.

## 4. Discussion

Although numerous classification systems for OPLL exist in academic circles, most of these systems describe and classify the shape of the ossification in order to provide more detailed information for diagnosis and optimize surgical guidance [23]. To our knowledge, no classification has addressed the key question that surrounds the underlying mechanism of ectopic ossification in OPLL. The present study assessed the morphological characteristics of new bone formation in OPLL and categorized them into plateau and hill shapes using spinal CT and utilizing the established morphological classification [16]. These two opposite appearance patterns have attracted recent attention by suggesting a difference in the ossification mechanism that might exist. Therefore, we further investigated its related factors to elucidate a part of OPLL ossification pathogenesis.

Previous studies have described two ways of osteophyte growth in DISH and AS, including one growth pattern that is more vertical and the other is more horizontal [11,12,13,14]. While horizontal growth is common in DISH, vertical growth is more prevalent in AS and indicates possible inflammatory pathogenesis. The present study suggests that spinal inflammation and plateau shape are linked in OPLL since the serum hs-CRP level in the plateau-shaped group was significantly higher than that in the hill-shaped group. This result is consistent with our previous work and other studies [11,12,13,14,15], demonstrating a similar mechanism between the plateau-shaped OPLL, the syndesmophyte AS, and the flat-type DISH. Thus, this tendency for more vertically oriented bone formations might indicate underlying inflammatory pathogenesis in both diseases.

Another hint that supports this theory is the pelvic involvement, mainly the SI joint. The SI joint degenerative changes are common in DISH, which is characterized by joint space narrowing, osteophytes, subchondral sclerosis, cysts, vacuum phenomena, and ankyloses [24,25]. Moreover, the ankylosed SI joint typically involves its upper non-synovial portion, characterized by para-articular bony bridging, while the lower synovial portion is spared [26,27]. Contrastingly, such a fusion in AS occurs due to synovitis with significant subchondral bone surface damage and erosions that are present in the intra-articular fusion [26]. Our recent study examined the SI joint variation in patients with OPLL using the whole spinal CT [20] and observed ankylosis not only in the entheseal sites but also in the synovial part of the SI joint with a high rate of intra-articular fusion. Thus, morphological similarities of the SI joint between OPLL and AS might be translated into pathogenic similarities at the onset of both conditions. The present study showed a significantly higher hs-CRP level in patients with SI joint intra-articular fusion compared to patients with anterior para-articular bridging. The elevation of the serum hs-CRP level has been shown to coincide with SI joint inflammation [28]. Additionally, increased CRP was frequently observed in patients with painful axial AS and was correlated with both disease activity and severity [29,30]. Interestingly, our current study showed a different SI joint fusion pattern between the plateau-shaped and hill-shaped groups. SI joint involvement in the plateau-shaped group was characterized by intra-articular fusion rather than anterior bridging, which was mainly identified in the hill-shaped group. These results, once again, strongly suggested that the different underlying mechanisms might partially affect the OPLL ossification process, as well as the phenotype of SI joint ankylosis. Thus, the associations between the SI joint intra-articular fusion, hs-CRP levels, and plateau-shaped bone formation are in concert with our hypothesis that there is some form of an inflammatory basis for the progressive bone formation in the spine and SI joint in OPLL.

The analysis of the ossified lesion characteristics revealed that the plateau shape was highly associated with continuous and segmental types, whereas the hill shape more frequently occurred in patients with mixed and localized types. Additionally, the maximum thickness of the ossified PLL in the plateau-shaped group was prevalent at the vertebral segment and largest at the C2 level. Meanwhile, the maximum thickness was mainly observed in the intervertebral segment in the hill-shaped group, which is a site subject to degeneration. Moreover, the thickness and OR of the hill shape were significantly higher, and the SAC was significantly lower compared to that of the plateau shape. The mechanical stress has been believed to be involved in the formation and progression of OPLL [31], and the dynamic motion could lead to biochemical responses by inducing osteogenic differentiation in the ligament cells [32,33]. Consequently, ossifications actively grew at intervertebral levels to bridge gaps and stabilize the mobile segments. As previously reported, the jaggy-typed DISH is predominant in the T8/9 level, where the natural kyphosis curvature begins, indicating a relationship with mechanical stress due to spinal movement [15]. Moreover, hs-CRP levels were lower in the hill-shaped group. This result and the distribution tendency suggest that the degeneration process rather than the inflammation pathogenesis is related to the onset of the hill shape in OPLL.

Classic degenerative spinal changes are also related to disc degeneration. Patients with OPLL comorbid with intervertebral disc degeneration (IVDD) are commonly observed in clinical practice. Luo et al. found a significant correlation between the ossification thickness and IVDD at corresponding levels in patients with OPLL [34]. Hanakita et al. suggested that growth factor secretion by a degenerated disc and mechanical stress imposed on ligaments can promote the ossification of OPLL [35]. Other histological studies of the tip-toe walking mouse identified that the chondrocytes of the intervertebral disc herniation could migrate to ligaments and continue to proliferate, causing a PLL transformation into the cartilaginous ossification to reinforce the vertebral column [36,37]. Thus, the predominant occurrence of the maximum thickness level of the hill shape in the intervertebral segment might be verified by the IVDD. However, because the IVDD in OPLL is usually mild and the IVDD was not evaluated in the current study, this issue remains controversial.

The current study had several limitations. First, this was a retrospective and cross-sectional study in a single institute with a relatively small number of patients. Second, serum hs-CRP levels were measured at only one point, with inadequate follow-up. A prospective study with a larger sample size, frequent observation periods, and multiple time-point evaluations of hs-CRP levels might be needed to verify our findings. Third, whether local inflammation occurs at the site of OPLL or not is not evident since we could not exclude all the patients with numerous factors other than cardiovascular diseases that might also affect the hs-CRP elevation, as well as because we could not evaluate the specific inflammatory factors from the ossified OPLL lesions and the surrounding tissues. Further studies are planned to explore this aspect in the near future.

## 5. Conclusions

In summary, the associations between the elevation of serum hs-CRP levels the occurrences of plateau-shaped ossified lesions and SI joint intra-articular fusion found in the current study suggest a possible inflammation mechanism for the progressive bone formation in the spine and SI joint in some cases of OPLL. However, all identified associations were weak, and thus, the role of inflammation in new bone formation in OPLL needs to be further investigated.

## Figures and Tables

**Figure 1 jcm-12-00349-f001:**
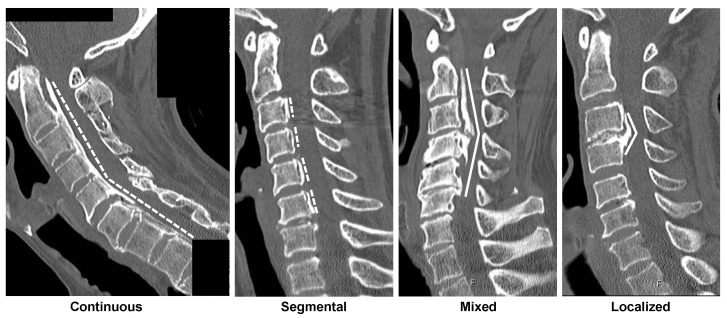
Representative OPLL type and OPLL shape. Plateau shapes are marked with dashed lines and indicate the relatively narrow spinal canal without any localized massive ossification. Hill shapes are marked with solid lines, defined by a massive beak-shaped ossification localized to certain levels [16]. The radiographs display (from left to right) plateau-shaped ossification in the continuous type OPLL, plateau-shaped segmental type OPLL, hill-shaped ossification in a beak configuration, and hill-shaped circumscribed type OPLL. OPLL, ossification of the posterior longitudinal ligament.

**Figure 2 jcm-12-00349-f002:**
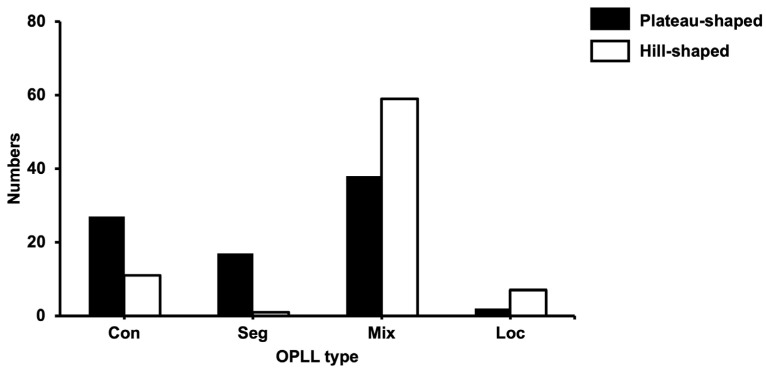
The relationship between OPLL type and OPLL shape. OPLL, ossification of the posterior longitudinal ligament; con, continuous; seg, segmental; mix, mixed; loc, localized.

**Figure 3 jcm-12-00349-f003:**
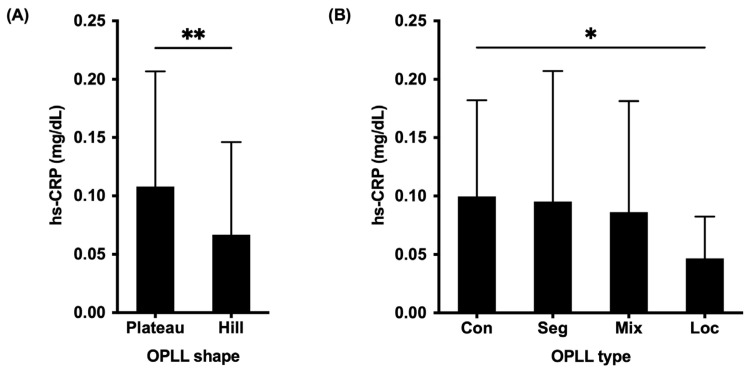
Comparison of serum high-sensitivity C-reactive protein levels between the OPLL types (**A**) and OPLL shapes (**B**). OPLL, ossification of the posterior longitudinal ligament; hs-CRP, high-sensitivity C-reactive protein; con, continuous; seg, segmental; mix, mixed; loc, localized; *, *p* < 0.05; **, *p* < 0.01.

**Figure 4 jcm-12-00349-f004:**
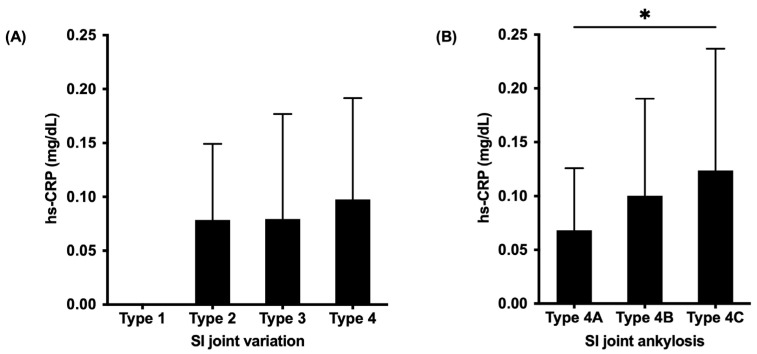
Comparison of serum high-sensitivity C-reactive protein levels between SI joint types (**A**) and subtypes (**B**) classifications [19,20]. SI, sacroiliac; hs-CRP, high-sensitivity C-reactive protein; *, *p* < 0.05.

**Table 1 jcm-12-00349-t001:** Demographic data and ossified lesion characteristics of patients with OPLL.

	Plateau-Shaped	Hill-Shaped	*p*-Value
Number of patients, n	84	78	-
Sex, n (%)			
Male	51 (60.7)	50 (64.1)	0.75
Female	33 (39.3)	28 (35.9)	
Age, year, mean ± SD	67.7 ± 10.4	67.2 ± 10.0	0.79
BMI, kg/m^2^, mean ± SD	24.9 ± 4.0	25.5 ± 4.8	0.48
Max ossified PLL thickness level, n (%)			
C2	13 (15.5%)	1 (1.3%)	0.011
C2/3	10 (11.9%)	7 (9.0%)	
C3	6 (7.1%)	3 (3.8%)	
C3/4	11 (13.1%)	18 (23.1%)	
C4	9 (10.7%)	10 (12.8%)	
C4/5	3 (3.6%)	11 (14.1%)	
C5	9 (10.7%	7 (9.0%)	
C5/6	4 (4.8%)	8 (10.3%)	
C6	8 (9.5%)	5 (6.4%)	
C6/7	6 (7.1%)	7 (9.0%)	
C7	5 (6.0%)	1 (1.3%)	
Vertebral	50 (59.5%)	27 (34.6%)	0.002
Inter-vertebral	34 (40.5%)	51 (65.4%)	
Max ossified PLL thickness, mm, mean ± SD	0.55 ± 0.17	0.67 ± 0.17	<0.0001
C2	0.64 ± 0.11	0.60	0.003
C2/3	0.62 ± 0.17	0.63 ± 0.13	
C3	0.35 ± 0.10	0.49 ± 0.06	
C3/4	0.63 ± 0.15	0.69 ± 0.17	
C4	0.53 ± 0.16	0.67 ± 0.16	
C4/5	0.61 ± 0.04	0.73 ± 0.13	
C5	0.43 ± 0.12	0.62 ± 0.18	
C5/6	0.53 ± 0.21	0.69 ± 0.18	
C6	0.44 ± 0.14	0.58 ± 0.20	
C6/7	0.63 ± 0.15	0.73 ± 0.26	
C7	0.62 ± 0.17	0.60	
SAC, mm, mean ± SD	0.80 ± 0.20	0.66 ± 0.28	0.002
Occupying ratio, %, mean ± SD	40.9 ± 10.4	51.5 ± 14.4	<0.0001
≥60%, n (%)	2 (2.4%)	22 (28.2%)	<0.0001
<60%, n (%)	82 (97.6%)	56 (71.8%)	

OPLL, ossification of the posterior longitudinal ligament; BMI, body mass index; PLL, posterior longitudinal ligament; SAC, space available for the cord.

**Table 2 jcm-12-00349-t002:** Occurrences of OPLL shapes and SI joint variation.

	Plateau-Shaped	Hill-Shaped	*p*-Value
**SI joint variation**	N = 84	N = 78	-
Type 1, n (%)	0 (0%)	0 (0%)	0.27
Type 2, n (%)	12 (14.3%)	13 (16.7%)	
Type 3, n (%)	35 (41.7%)	23 (29.5%)	
Type 4, n (%)	37 (44.0%)	42 (53.8%)	
**SI joint ankylosis**	N = 37	N = 42	-
Type 4A (Anterior), n (%)	10 (27.0%)	24 (57.1%)	0.03
Type 4B (Posterior), n (%)	4 (10.8%)	3 (7.1%)	
Type 4C (Intra-articular), n (%)	23 (62.2%)	15 (35.7%)	

OPLL, ossification of the posterior longitudinal ligament; SI, sacroiliac.

## Data Availability

The data underlying this article cannot be shared publicly due to the privacy of individuals that participated in the study. The data will be shared upon reasonable request with the corresponding author.

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
