# Peer review of "Association of Inflammation, Ectopic Bone Formation, and Sacroiliac Joint Variation in Ossification of the Posterior Longitudinal Ligament"

_jcm, 2023, doi:10.3390/jcm12010349_

Round 1
Reviewer 1 Report
The iconographic documentation is insufficient and incomplete.
-Figure 1. pag 2, line 111, 112, 113, 114: it can be better described, in the caption there are no references to the definition of plateau-shaped and hill-shaped.
-We note the lack of a Figure 2, with CT and Rx images with the 4 types of ossification of the SI.
Reviewer 2 Report
The issue of concern is that there are numerous factors not accounted for in this analysis that may attribute to the findings presented. Additionally, the sample size is quite low to assess such a broad relationship.
The conclusion of causation is not able to be answered by this study design.
Reviewer 3 Report
This is an interesting manuscript evaluating potential mechanistic causes of OPLL. The authors hypothesize that plateau shaped OPLL may be linked to a more inflammatory state than “hill” type OPLL and link these types of OPLL to SI joint inflammation and fusion. This was a well written manuscript and it was very informative. The introduction is excellent and provides the reader with an appropriate overview of the recent updates on OPLL. The methodology was well performed and I really have no recommendations for improvement. I do think it would be nice to know if each patient presented with myelopathy and if the authors have JOA scores I believe it may be of interest to surgeons. It seems that even though the plateau type is more inflammatory it would be less likely to cause severe myelopathy which is an interesting finding. The results, discussion and conclusion are well done. Overall I would like to congratulate the authors on an interesting well performed study.
Round 2
Reviewer 2 Report
The current article has limitations in its study design. The sample size and statistical analysis do not have sufficient confidence in determining the stated relationship. There are too many genetic, demographic, and patient and surgical factors that may contribute to the relationship.